# Application of Pineapple Waste to the Removal of Toxic Contaminants: A Review

**DOI:** 10.3390/toxics10100561

**Published:** 2022-09-26

**Authors:** Bienvenu Gael Fouda-Mbanga, Zikhona Tywabi-Ngeva

**Affiliations:** Department of Chemistry, Center for Rubber Science and Technology, Nelson Mandela University, Gqeberha 6031, South Africa

**Keywords:** pineapple, pollutants, environmental waste, nanomaterials

## Abstract

The presence of pollutants in large swaths of water is among the most pressing environmental issues of our time. This is mainly due to the inappropriate disposal of industrial sewerage into nearby water supplies and the production of a broad range of potentially hazardous contaminants. Pineapple is a fruit mainly grown in tropical regions. Refuse production begins with the collection of raw materials and continues prior to being refined. Pineapple processing industries generate waste (peel, core, pomace, and crown) that is high in bioactive compounds. The byproducts often include more valuable compounds with greater nutritional and therapeutic value than the final product. This review focuses on the application of pineapple and components, adsorbent synthesized from pineapple for the removal of pollutants.

## 1. Introduction

Water is regarded by far the most significant source of life that is recyclable, in which surface and groundwater play critical roles in agriculture, livestock production, hydropower generation, etc. [1,2]. The pace of demographic growth around the world is rising daily. This increase in population comes eventually with several environmental apprehensions [3,4]. Water pollution is among the challenging concerns that humanity has ever dealt with. Water quality is utterly important to people’s wellbeing. According to the World Health Organization (WHO), approximately 829,000 deaths each year are caused by the poor quality of water [5].

Several pollutants promote the pollution of water among which we have heavy metals, dyes, fertilizers, pesticides, and pharmaceuticals. The major source of these pollutants is mining, electroplating plants, welding, metal finishing, and alloy production [6]. Concerns about the presence and contamination of heavy metals in our water resources have risen significantly in the past years [7,8]. At relatively low concentrations, elements like mercury (Hg) and cadmium (Cd) are toxic to humans. Silver (Ag), chromium (Cr), lead (Pb), copper (Cu), and zinc (Zn) are also harmful to humans, albeit at much higher concentrations than those required for cadmium or mercury toxicity [9].

It is essential to eliminate toxic metals from aqueous media in places such as South Africa where drinking water is scarce. Toxicity caused by toxic metals like cadmium (Cd), lead (Pb), and manganese (Mn) can have serious health effects. Human activities have contributed greatly to the contamination of aqueous systems, especially with toxic metals; therefore, we need efficient and cost-effective methods to remove them. Physico-chemical techniques have been applied for centuries to eliminate metals from aqueous streams, including chemical precipitation, ion exchange, and adsorption. Despite the risks of incomplete removal, such techniques need the use of a substantial number of chemicals at great expense, especially when contaminant concentrations are in the range of 10 to 100 mg/L [10]. They are also time-consuming and generate additional waste [11,12]. Researchers have reported advances in the removal of heavy metals from aqueous media by using microorganisms, with special attention on the use of fungal species in either batch or continuous mode. Due to its flexibility in design, lack of harmful substance generation, and simplicity of adsorbent regeneration after an application, adsorption has been widely accepted as a highly effective and widely used method for water and wastewater treatment [13,14]. Water treatment and beneficial re-use can be accomplished with adsorption applications [10]. Various adsorbents such as metallic and non-metallic, graphene-based, organometallic systems (MOFs), polymer-based carbon nanotubes (CNTs), and basically carbon-based activated carbon have been applied to handle wastewater [15,16,17,18]. However, because of their complicated preparation, high cost, unstable nature, low hydraulic conductivity, and other factors, various adsorbents have limited application [19,20]. On the other hand, activated carbons demonstrated encouraging findings in wastewater purification because they are cost-effective, easy to prepare, and possess large specific surface area. Carbon nanomaterials are excellent transporters for isolating inorganic and organic contaminants by sorption due to their significant adsorption capacity and remarkable selectivity. Carbon-based adsorbents for pollutant removal have been a focus of research in order to improve and optimize the preparation process. Undoubtedly, interactions between carbon functional groups are critical in the elimination of different contaminants from aqueous solutions [21,22].

Agro-wastes and other plant wastes have recently gained attention as sorbents for toxic metals in aqueous systems due to their ability to adsorb toxic metals from aqueous systems [23,24]. In addition to being easily accessible, inexpensive, recyclable, sludge-free, and requiring very little initial capital and land investment, agro-wastes also has several other advantages. If agricultural and industrial wastes are not properly recycled and reprocessed, they pose a significant burden on the environment. Agricultural waste comes from a variety of sources derivatives of agricultural processing, like peels, pits, shells, leaves, and so on, and are abundant natural resources that should be extensively considered for the production of value-added materials [25,26].

## 2. Pineapple Waste Application as an Economic-Development Means of Waste Management

Pineapples are perennial herbaceous plant that are grown alongside coastlines and in tropical areas such as Indonesia, China, Malaysia, South Africa, Costa Rica, Nigeria, Philippines, Thailand [27]. Pineapples fruits are sprouted on about 2,250,000 acres of land in India, for example. Based on a 2006 Taiwan Council of Agriculture report, Taiwan’s total pineapple cultivation area is 12,225 acres, and 439,872 tons of pineapples are harvested every year.

Pineapple production in South Africa is concentrated in two regions: Eastern Cape Province (Bathurst Region), which accounts for 66% of the total production, and KwaZulu-Natal Province (Hluhluwe Region), which accounts for 33%. According to Statistics South Africa [28,29], pineapple production is also limited in Mpumalanga and Limpopo provinces, accounting for less than 1% of total production. About 80% of South Africa’s pineapple production is used for processing, with the remainder being sold in the local fresh fruit market. South Africa currently exports small quantities of pineapples to the United States as the subtropical fruit is eligible for duty-free access under the African Growth and Opportunity Act [30].

The first leaf sprout is soft. Gradually, the leaves stiffen, turning sword-shaped and spiraling around the fruit [26,31]. It is the most popular edible member of the bromeliad family, and its juice ranks third in the world after orange and apple juice [32]. In addition to agricultural uses such as nutraceuticals, this plant has a wide range of pharmacological properties, namely antibacterial [33], antihyperlipidemic, antidiabetic, and antitumor properties [34,35]. Kalpana et al. [36] investigated the antidiabetic and antioxidant features of pineapple leaves in streptozotocin (STZ)-induced experimental rats. Leaves are the waste biomass of pineapple fruits and are sometimes utilized as a source of natural fibers.

### 2.1. Pineapple Waste as an Environmental Threat

Disposal of pineapples consists of the remains of the peel, pulp, stems, and leaves left over from the processing of pineapples. Pineapple leaf fibers are principally made up of holocellulose and lignin, with trace amounts of ash [37]. Due to the fact that pineapple plant leaves give rise to solid, fine, silky fibers, some research has been directed towards enhancing the application of this natural bio-adsorbent in short and fiber-reinforced rubber composites [38,39].

The quantity of waste produced by pineapple leaf waste is concerning, with approximately 20,000–25,000 tons per acre left over after the harvesting process [40]. It is mainly caused by lack of proper fresh fruit management, insufficient transportation options or insufficient storage.

The pineapple industry’s wastewater generates significant amounts of Biochemical Oxygen Demand (BOD), Chemical Oxygen Demand (COD), and Suspended Solids. High COD contents in wastewater are harmful to biological life and will have an impact on the aquatic environment. The pineapple refining business used a lot of water at almost every step of the production process. It produces a huge amount of solid and liquid waste as a consequence of these activities. Unrefined pineapple business effluent comprise greater levels of carbohydrates such as sucrose, glucose, and fructose, which may cause environmental issues when released into rivers [41].

### 2.2. Pineapple Adsorbent and Composites Used for the Elimination of Inorganic and Organic Pollutants

Unprocessed and activated pineapple leaf demonstrated a positive removal ability for distinctive type of pollutants. Chowdhury and co-workers used pineapple leaf powder (PLP) to remove basic green 4, a cationic dye from aqueous solution. Their findings demonstrated that PLP adsorbent demonstrated higher uptake capacity than other reported adsorbents in literature [42]. Ponou et al. [43] used carbonize pineapple leaves to eliminate chromium (Cr VI) anions in aqueous solution. The mechanism was found to be chemisorption, agreeing with a heat-absorbing reaction, and the system is well described by a pseudo-second-order kinetic model. The Langmuir and Freundlich models were applied to describe these uptake isotherms. Magnetic activated carbon (MAC) was prepared from pineapples’ crown leaves and utilized to remove methyl violet dye from aqueous solution. It was discovered that methyl violet dye onto MAC was best described by the Redlich–Peterson isotherm model [44]. Mohammed et al. [45] used pineapple waste as agricultural adsorbent for the removal of safranin-O from water. They indicated that adsorbent dose, pH, and contact time were all varied. Dye uptake was initially very rapid but gradually slowed, indicating penetration into the interior of the adsorbent particles. It was discovered that acidic pH was more conducive to adsorption. The highest uptake capacity was attained after 90 min of contact between the adsorbate and the adsorbent at 29 °C. The results obtained fit the Freundlich and Langmuir models; the Freundlich model described steady-state dye uptake better than the Langmuir model. In another study, Veeramalai et al. [46] used process pyrolysis to change pineapple waste biomass (PWB) into useful adsorbents including biochar (BC) and activated carbon (AC) for lipase immobilization and RBBR (remazol brilliant blue R) dye adsorption. The performance of BC in lipase immobilization and RBBR dye adsorption was studied by using different parameters such as initial concentration, physical size of PWB (grounded and non-grounded pineapple waste (PW) biomass), and physical size of BC (crushed and non-crushed BC). The results of generated BC were evaluated by comparing to those of commercial AC. The highest amount of protein adsorbed during immobilization was obtained at F500 1 h PWbB (92.99 percent). Meanwhile, the best dye removal was obtained at F 600.5 h PWbB. (83.59 percent). Yamuna and Kamaraj [47] prepared pineapple peel waste activated carbon for the effective removal of methylene blue from aqueous solution. In their investigation, chemical modification was utilized to enhance adsorption capacity. The outcomes showed that as the amount of adsorbent increased, so did the degree of adsorption, and equilibrium adsorption was reached in 30 min [47]. Powder from pineapple crown leaves was used as an adsorbent for the removal of crystal violet from aqueous solution as stated by Nieva et al. [48]. In their investigation, hydroxyl and carboxyl functional groups were responsible for the surface bonding with the crystal violet. Potassium ion was the main element present on the adsorbent justifying the affinity of the adsorbent with the crystal violet. The maximum adsorption capacity was 6.4935 mg/g and the adsorption was best demonstrated by the Langmuir isotherm [48]. Dai et al. [49] reported an easy method to synthesize pineapple peel cellulose/magnetic diatomite hydrogels in an environmentally benign ionic liquid, 1-butyl-3-methylimidazolium chloride for the removal of methylene blue. The methylene blue adsorption process was fast (equilibrium reached in 30 min) and fit well into the pseudo-secondary reaction rate model and the Langmuir isotherm model. The maximum adsorption volume calculated from the Langmuir isotherm model was 101.94 mg/g, which was clearly higher than the hydrogel (75.87 mg/g) prepared without the addition of magnetic diatomite. In addition, the prepared hydrogel showed good stability and reusability to MB adsorption [49]. Beltrame et al. [50] synthesized mesoporous activated carbons from pineapple plant leaves to remove caffeine. The procedure was achieved using phosphoric acid as an activating agent and slow pyrolysis in an N_2_ atmosphere. Activated carbon fibers exhibited fibrous characteristics with a preponderance of acid groups on their surface. Adsorption studies revealed that the pseudo-second order kinetic and Langmuir isotherm models fit the experimental data the best. The capacity of a single layer adsorption was found to be 155.50 mg g^−1^ [50]. Table 1 below presents various adsorbents prepared from pineapple leaf for the removal of dye in aqueous solution.

Table 1 above shows that pineapples and derivatives when applied as adsorbent are efficient in the removal of dyes from water. Furthermore, as expected, when these adsorbents are modified or doped with activated carbon or biomolecules, the adsorption capacity increases significantly. Additionally, the table presents different methods that were used to prepare the adsorbents. The advantages and disadvantages of these methods are illustrated in Figure 1 below:

### 2.3. Pineapples Applied as an Adsorbent for the Removal of Heavy Metals

Several studies in which pineapple leaves and stems were used as potential adsorbents or precursors are reported in literature. Amsyar Ayob et al. [63] used pineapple waste for the removal of Pb^2+^ from synthetic wastewater. In their work, it was found that treated pineapple waste with NaOH yielded a higher adsorption efficiency than the untreated pineapple waste. The maximum adsorption capacity happened at pH 2. Rergnaporn and Nanthaya [63] investigated the efficiency of sodium hydroxide-treated pineapple waste for the removal of lead and cadmium from aqueous solution. In their works, it was discovered that adsorbents derived from various pineapple wastes treated with sodium hydroxide have relatively high cadmium and lead ion adsorption efficiency above 95% than untreated ones whose efficiency was in the range of 50–100%. For both NaOH-treated and untreated wastes, Pb^2+^ adsorption efficiency was greater than Cd^2+^ adsorption efficiency. The adsorption efficiencies for Pb^2+^ and Cd^2+^ ions in untreated and treated fruit waste are lower than in leaf, stem, and mixed waste [64]. Mopoung and Buntern [64] investigated the modification of pineapple fiber waste carbon with potassium permanganate for the removal of ferric ion from aqueous solution. The findings demonstrate that MnO_2_ deposited on the surface of KMnO_4_ functionalized waste carbon in a heterogeneous manner. On the surface of KMnO_4_ functionalized carbon, the OH, CO, and MnO groups are the most prominent functional groups. Adsorption of Fe^3+^ on functionalized waste carbon achieves equilibrium in 60 min [65]. Table 2 below illustrates some studies in which pineapple and derivatives were used with the relevant conditions for the removal of heavy metals.

Table 2 demonstrated non-modified adsorbent (pineapple leaf powder, natural pineapple plant stem) yielded lower adsorption capacity compared to modified adsorbents. Figure 2 below outlines (A) the adsorption mechanisms of Cr(VI) on APF-PEI adsorbent; (B) adsorption of Cr(VI) and Cr(III) ions from aqueous solution by PCL; (C) the removal of heavy metals using modified celluloses prepared from pineapple leaf fiber.

### 2.4. Application of Pineapple Wastes

Pineapple wastes have been applied in different applications, namely, energy generation or carbon source, antioxidant, pharmaceutical and food industry, citric and lactic acid production, ethanol production, vinegar production, fiber production, etc. [75,76,77].

#### 2.4.1. Energy Production and as a Carbon Source

The possibility of producing energy from pineapple waste was investigated in the past. Oranusi et al. [74] published data on biogas production (71 percent CH_4_, 18 percent CO_2_, 7.0 percent N_2_, 1.5 percent H_2_, 1.5 percent H_2_S, 1 percent O_2_) through the co-digestion of pineapple peels with food waste (1:1) with cattle rumen as inoculums [78]. It has also been found that pineapple biomass produces more biogas and has a faster activity rate than watermelon biomass. This is due to the high fermentable sugar content of pineapple biomass, which allows hydrolytic bacteria to act quickly, as opposed to watermelon biomass, which has more fibrous tissues [79]. Biomethaneization is a process known as anaerobic digestion, which converts organic matter in waste into methane and fertilizer through the action of microorganisms in the absence of air. Biomethaneation is the anaerobic digestion of biodegradable organic waste in a closed space under controlled temperature, moisture, pH, and other parameters. This is a decomposition system that anaerobically decomposes a mass of waste according to the characteristics of the waste to produce biogas, which is mainly composed of methane and carbon dioxide [80]. Syaliza et al. [77] reported the production of biogas from banana and pineapple peels using anaerobic digestion. For 5 days, the investigation was carried out in a mesophilic environment (37 °C), and the optimized pH value was between 6.8 and 7.2. The findings showed a total solid content of 17–19%, a volatile solid content of 85–95%, and a pH range of 3.3–6.7%. While total biogas results revealed that pineapple peel had a higher value (571.4 mL) than banana peel (386.6 mL) [81]. In a similar study, Suphang et al. [78] utilized pineapple peel solid waste to successfully generate methane gas at a concentration of 48% at 20 days. Pineapple peels were used as a potential carbon source for fermentation medium in combination with municipal wastewater for biodiesel production with Candida tropicalis [82]. Pineapple waste has been used as a carbon precursor in the production of hydrogen gas from municipal sewage sludge and the production of cellulose by Acetobacter xylinum [83,84].

#### 2.4.2. Antioxidant Activity

According to researchers, pineapple peel is a potential source of bioactive compounds such as vitamin C, carotenoid, phenolic compounds, and flavonoids, and these compounds have antioxidant activity as well as other biological activities [85,86,87]. According to certain studies, fruit residues, including pineapple, have the same antioxidant activity as the fruit pulp. Even though these byproducts are disposed, they could be used as a source of polyphenols and natural antioxidants as an alternative. Francisco Segovia Gómez and María Pilar Almajano Pablos [88] reported that pineapple waste extracts reduced oxidation products by 59 percent in emulsions and 91 percent in muffins. Saraswaty et al. [85] presented an alternative approach of using both fresh and dried pineapple peel wastes as an antioxidant. In their work, they used a mixture of pineapple peel waste with ethanol and water at different concentration. The findings revealed that the antioxidant activity of extracts from dried and fresh pineapple peel waste was in the scope of 0.8 ± 0.05 to 1.3 ± 0.09 mg·mL^−1^ and 0.25 ± 0.01 to 0.59 ± 0.01 mg·mL^−1^, respectively [89]. Other studies looked at the anti-inflammatory and anti-diabetic properties of pineapple stem waste. Investigation on phytochemicals derived from pineapple peel and leaf has also revealed a high antioxidant activity with a high concentration of phenolic compounds [90,91,92]. The leaf also contains a high concentration of phytosterols, particularly beta-sitosterol, stigmasterol, and campesterol. Furthermore, the most phenolic from pineapple peel was extracted in 30 min using 75% ethanol at 75 °C (unpublished data). De Oliveira et al. [89] focus on the antioxidant potential of pineapples. In their work, they investigated the total phenol content and the antioxidant activity of methanol extract of pineapples (including flesh, seeds, and skin from the local juice factory), using the DPPH (2,2′-diphenyl-2-picrylhydrazyl radical) and the activity of removing superoxide anions [93]. Yahiya et al. [90] extracted polyphenol from pineapple skin. The results showed that the polyphenol content in the extractors was the highest in methanol, corresponding to the highest antioxidant properties. Gallic acid, catechine, epicatechin, and pherulic acid were found to be the main polyphenyls in the pineapple skin. Polyphene interactions did not reveal synergetic effects [94]. Using both antioxidant activity analyses, addictive effects have been detected in the combination of pherulic acid epicatechine and pherulic acid halonic acid [95].

#### 2.4.3. Anticancer and Antibacterial Activity

Hakal et al. [77] reported that ferulic acid can be extracted from pineapple and used as an antioxidant and anticancer. Ferulic acid is an organic compound that is hydroxycinnamic acid. This is the high amount of phenolic photochemicals found in pineapple skins. It is associated with transcinnamic acid and contributes to the plant’s antibacterial, anticancer, and antioxidant properties [95].

Pineapple waste has the potential to be a source of new medicines to treat food contamination and other infectious bacteria. Lubaina et al. [96] detailed their study in which the antibacterial activity of pineapple peel ethyl acetate extract was effective against both Gram-positive and Gram-negative bacterial strains. Their results demonstrated that pineapple peel has the potential to be a useful source of antibiotics and has medicinal and industrial applications [97,98]. Niramol et al. [95] reported that pineapple peel extract could be a potential source of antimicrobial agent and may be applied for food application. The outcomes of their investigation revealed that Gram-positive bacteria were found to be more susceptible than Gram-negative bacteria. The most resistant bacteria, according to the broth dilution assay, was B. cereus. Except for B. cereus, the concentration that had a lethal effect on the bacteria tested was 0.0675 g/mL [99].

The presence of polyphenols, flavonoids, saponins, and other secondary metabolites in pineapple extracts may explain its antibacterial activity. Flavonoids and polyphenols have a greater inhibitory effect on Gram-positive bacteria. Both are phenolic compounds with polar properties that act primarily in the peptidoglycan layer of Gram-positive bacteria rather than the non-polar lipid layer. As the majority of plant phenolic compounds are non-toxic to humans, they could be used to inhibit the growth of many foodborne and food spoilage microorganisms [100]. The antimicrobial properties of the different extracts from pineapple peel have recently attracted the attentions of researchers due to their potential application as natural additives, which stems from a growing trend to reinstate synthetic antimicrobials with natural antimicrobials [98].

#### 2.4.4. Pharmaceutical and Food Industry

To explore the application of pineapple waste in the pharmaceutical and food industry, researchers extracted from pineapple waste a potential source of protease known as Bromelain [101]. Although Bromelain extraction and purification methods have been investigated, there are some limitations due to the laboratory-scale nature of most methodologies, which raises operating costs [102].

Because of the wide range of applications for bromelain enzyme, commercial bromelain must be highly pure [103]. Bromelain has been used in the food industry for meat tenderization, brewing, baking [104,105], apple juice [106], browning prevention, beer clarification, and as a food supplement [107]. Bromelain promotes meat softening by breaking down fibrous material during the tenderization process. Bromelain improves dough relaxation in the baking industry, allowing it to rise evenly and produce hypoallergenic flour suitable for wheat-allergic patients. Bromelain has been used in this industry because its ideal temperature range of 50–70 °C is appropriate for food processing. Bromelain has numerous applications as an active ingredient in tooth-whitening dentifrices and skin products to treat acne, wrinkles, and dry skin, as well as to reduce post-injection bruising and swelling [108]. It is used as an active ingredient to provide mild peeling effects [107] and as a cleansing agent [109]. This enzyme is widely used in the pharmaceutical industry as a drug for the treatment of inflammatory ailments, intestinal disorders, blood-coagulation related diseases, improved antibiotic absorption wound debridement agent, fibrinolytic agent [101,109], oral treatment for third degree burns, therapeutic application for antibodies [110], and mucolytic action [111]. Bromelain has gained popularity as an herbal medicine due to its lack of side effects and effectiveness after oral administration. It is available as tablets and capsules. In fact, Bromelain is only poorly absorbed when taken orally, resulting in plasma levels of less than 10 ng/mL in humans given 4 g per day [112].

#### 2.4.5. Production of Ethanol

Pineapple skin waste contains enough carbohydrates and reducing sugars to be converted into bioethanol via fermentation and distillation. Pineapple peelings are also used as a biomass in bioethanol production due to their high carbohydrate content, which can be converted into fermentable sugar. Carbohydrates in biomass are represented as total structural carbohydrates, which account for 37% of dry pineapple waste and are thus regarded the primary component of dry pineapple waste. The component of pineapple peelings that can be converted to ethanol is known as ethanol extractive. At 22.2 percent, ethanol extractive covers a significant amount of dry pineapple waste. In addition to ethanol extractive and total structural carbohydrates, dry pineapple waste contains 7.5 percent acid-insoluble lignin, 0.96 percent acid-soluble lignin, 5.4 percent ash, 27.14 percent protein, and acetic acid. Antonio et al. (2015) investigated how bioethanol can be manufactured from pineapple peelings using saccharomyces cerevisiae as fermenting yeast with the focus on fermentation pH. Their findings revealed that ethanol was produced at a maximum concentration of 9.13% at pH 5.5 [113]. In a study done by Tropea et al. [114], after 30 h of simultaneous saccharification and fermentation, the highest ethanol yield was achieved, reaching up to 3.9 percent (*v*/*v*), corresponding to 96 percent of the theoretical yield.

#### 2.4.6. Production of Vinegar

Praveena and Estherlydia [115] reported on the conversion of pineapple peel to vinegar via simultaneous fermentation with Saccharomyces boulardii and Acetobacter, and the resulting vinegar was shown to have phytochemical and antioxidant properties. A comparison of the phytochemical screening and antioxidant capacity of vinegar revealed that vinegar made from pineapple peel had higher antioxidant activity (2077 mg acetate equivalent/100 mL) than vinegar made from other fruit wastes. This product, as opposed to synthetically produced chemical vinegar, can be produced and marketed in large quantities due to its therapeutic effects and environmental friendliness. Tanamool et al. [116] reported a potentially newly isolated thermotolerant acetic acid bacteria (TH-AAB) with ethanol and acetic acid tolerance that was found to be very effective in the production of vinegar from pineapple peels and as an alternative, low-cost raw material via simultaneous vinegar fermentation (SVF). Their findings indicated that at an initial pH of 5.5, using the whole pineapple peel with the addition of diammonium phosphate (DAP) and MgSO_4_ produced slightly more acetic acid than juicing.

### 2.5. Nutritional and Health Benefits

Pineapple is one of the juiciest fruits and an absolute delight. It can be served with whipped cream, pudding or just as is. Both delicious and refreshing, pineapple juice is one of people’s favorite drinks during the summer. The best thing about a pineapple is that it is rich in nutrients and beneficial enzymes that not only ensure a healthy body but also a glowing complexion.

Pineapples are reputed to be very efficient in treating stultification and improper intestinal movement because they are rich in fibers, so the intestinal movements are regular and light. For any kind of morning nausea, motion dizziness, or nausea, drink pineapple juice. It works effectively in the elimination of nausea and vomiting. It contains almost no fat and cholesterol, and is loaded with the basic nutrients and vitamins the body needs to grow and develop in general. Fresh pineapple juice can be used to relieve air people’s infections, ephtheria, and chest infestation. Not only does it contain enough vitamin C, but it also contains an enzyme called bromineline, known to melt and decapitate the mucus. Pineapple is effective in the elimination of intestinal worms and also maintains intestinal and kidney cleanness. It is effective in exorcising toxins from the body, thus maintaining a healthy metabolism. The pineapple contains a very high proportion of manganese, and one cup of pineapple juice is said to contain a good amount of pineapple. Table 3 below presents the nutritional value in 100 g of pineapple [117].

### 2.6. Challenge and Future Trends

Even though the pineapple sector has been a significant key in the economies of many countries, mainly subtropical regions, some problems emerged in recent years. Among which we have selling, the time allocated for harvesting pineapple fruit, cannot be used to sell. More time should be found for sales. The selling price of pineapple is sometimes inexpensive; handling, the processing methods are often simple and they must be enhanced, and crop diseases need solutions to control fruit diseases which need to be ameliorated for better production. These concerns must be addressed, otherwise the expansion of the pineapple in these countries would be hampered [118]. In a study done by William et al. [119], it was demonstrated that the sharp rise in temperature increases the likelihood of pineapple production.

Researchers must investigate various purifying methods, isolation, stabilization, and the implementation of Bromelain recovered from pineapple waste for industrial and medical applications. Enzyme enhancement must be unearthed, and its efficiency must be assessed in a variety of applications, not just medical and food industry applications. Using food waste to make wine and vinegar is a low-cost solution to the excessive usage of land and feed. It is critical to evaluate the effectiveness of pineapple byproducts in terms of consumer health, benefits, and industrial commercialization potential [120].

Pineapple cellulose has been shown to create bioplastics due to its low mechanical strengths, solubility, and structure cohesion. These products must also be well structured, engineered, and suitable for the best foods and their products. Bioplastic production is limited [116]. To minimize the cost of the enzymatic hydrolysis pre-treatment and purification procedure and achieve high output, pineapple waste must first undergo pre-treatment. Additionally, to meet the need for pineapple trash to be converted into biofuels, pineapple waste must first go through pre-treatment.

## 3. Conclusions

To summarize what has been said thus far, this study emphasized worldwide pineapple manufacturing, environmental pollution, and the ongoing transformation of pineapple waste into value added products. As per currently accessible pineapple data records, there is a need for pineapple fruits. An influx in world economic pineapple production has resulted in increased waste generation. It also contributes to the surge in waste at landfills and carbon emission gases.

Clean water is still a concern in many countries in the world and there is a need to tackle this scourge. Pineapples being one of the main economic riches of many tropical countries are also a source of water contamination in those countries. It is therefore important to develop approaches that will promote the use of the pineapple and components in different economic sectors to avoid pollution. This review focused on the nutritional, biological, and economic significance of industrial wastes generated during pineapple processing, and the adsorbents used from pineapple to remove pollutants from water. Furthermore, this study demonstrated that pineapple and based products may be used as energy production, antioxidant, pharmaceutical products etc. Different adsorbents were successfully developed from pineapple leaves for the removal of pollutants (organic and inorganic pollutants) for water purification. However, these adsorbents end up as pollutants once discharged in the environment. Therefore, it is also critical to re-use the adsorbents prepared from pineapples when removing pollutants from wastewater to avoid secondary pollution through different applications such as fertilizers, soil conditioning, latent fingerprints, gender-based violence tracking devices, etc.

## Figures and Tables

**Figure 1 toxics-10-00561-f001:**
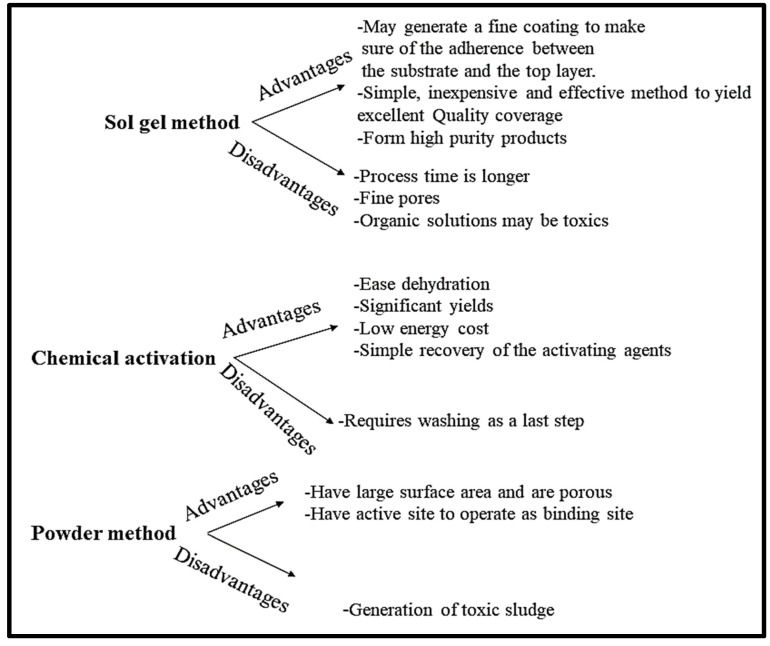
Advantages and drawbacks of powder, chemical activation, and sol gel methods [60,61,62].

**Figure 2 toxics-10-00561-f002:**
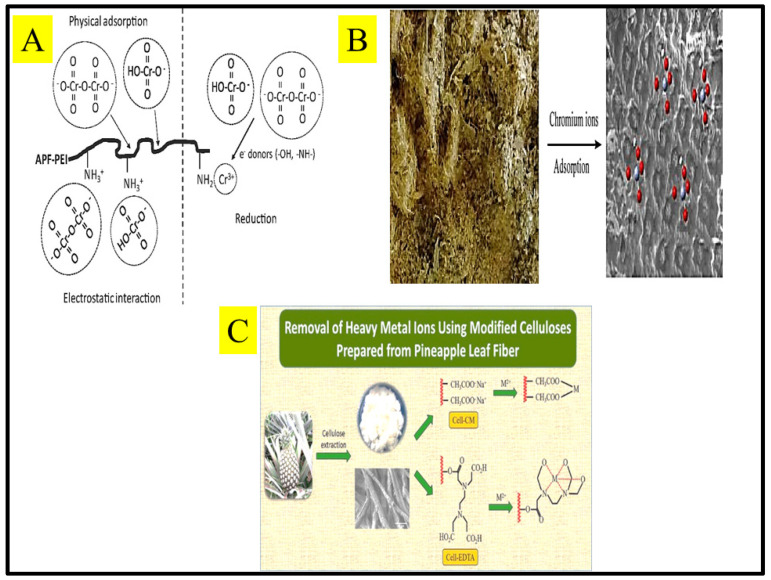
(**A**) the adsorption mechanisms of Cr(VI) on APF-PEI adsorbent; (**B**) Adsorption of Cr(VI) and Cr(III) ions from aqueous solution by PCL; (**C**) the removal of heavy metals using modified celluloses prepared from pineapple leaf fiber [67,68,71].

**Table 1 toxics-10-00561-t001:** Summary of various adsorbents prepared for the removal of dyes in aqueous solution.

Adsorbent	Preparation Method	Dye Removed	pH	Adsorption Capacity (mg g^−1^)	Isotherm	References
Titanium dioxide nano bio-adsorbent (TiO_2_L) based on *Ananas comosus* leaf extract	Sol gel	Victoria blue	6	83	Langmuir	[51]
Pineapple leaf fibers-cl-poly (acrylic acid-co-2-dimethyl amino ethyl acrylate)	Hydrogel	Methyl violet	3	625	Freundlich	[52]
Pineapple stem	Powder	Methylene blue	2–4	119.05	Langmuir	[53]
Activated carbon pineapple crown	Chemical activation using NaOH and pyrolysis	Methylene blue	6	292	Langmuir	[54]
Pineapple leaf	Powder	Remazol brilliant blue R	N/A	9.58	Langmuir	[55]
Pineapple bark	Powder	Congo red	9.8	N/A	N/A	[56]
Pineapple bark	Powder	Brilliant green	9.8	N/A	N/A	[56]
Pineapple bark	Powder	Methylene blue	9.8	N/A	N/A	[56]
Cellulose acetate from ananas cosmus leave	Ethanol and toluene (1:2)	Direct red	N/A	71.43	Freundlich	[57]
Pineapple peel	Powder	Eosin yellow	N/A	11.76	Langmuir	[58]
Pineapple leaf powder	Powder	Methylene blue	4	52.6	Langmuir	[59]
Surfactant-modified pineapple leaf powder	Hydrogel	Methylene blue	4	52.6	Langmuir	[59]
Pineapple leaf powder	Powder	Methylene orange	3	47.6	Langmuir	[59]
Surfactant-modified pineapple leaf powder	Hydrogel	Methylene orange	3	47.6	Langmuir	[59]

N/A: not available.

**Table 2 toxics-10-00561-t002:** Summary of some reported studies in which pineapple and components were used as adsorbents for the removal of heavy metals.

Adsorbents	Heavy Metal Adsorbed	pH	Maximum Adsorption Capacity (mg g^−1^)	Isotherm Representing Best the Adsorption	References
KMnO_4_ modified carbon from pineapple leaf fiber waste	Fe^3+^	4.7	25.25	Langmuir-Freundlich	[66]
APF	Cr (VI)	3	133	Langmuir	[67]
APF-PEI	Cr (VI)	3	222	Langmuir	[67]
PCL	Cr(VI)	1.5		Liu model	[68]
	Cr (III)	5			
APF-PEI	Cu^2+^	5	237	Langmuir	[69]
	Pb^2+^	5	165		
MPPF-DM70	Cu^2+^	5.5	65.98	Langmuir	[70]
	Cd^2+^	7.5	102.92		
	Pb^2+^	5.5	111.41		
Pineapple leaf fiber	Pb^2+^	6	63.92	Langmuir	[71]
	Cd^2+^	6	48.02		
Chemically oxidized pineapple fruit peel	Cd^2+^	4	42.10	Langmuir	[72]
	Pb^2+^		28.55		
Natural pineapple plant stem	Pb^2+^	5	14.25	Langmuir	[73]
Oxylic acid pineapple plant stem	Pb^2+^	4	30.47	Langmuir	[73]
DETA coated pineapple aerogel	Ni^2+^	N/A	49.00	Langmuir-Freundlich	[74]

APF: Alkaline pineapple leaf fiber; APF-PEI: polyethyleneimine-carbamate linked pineapple leaf fiber; PCL (Pineapple crown leaves); N/A: not available

**Table 3 toxics-10-00561-t003:** Nutritional value in 100 g of pineapple.

Calcium	Energy	Carbohydrates	Dietary Fiber	Iron	Magnesium	Protein	Phosphorous
16 mg	52 Calories	13.7 gm	1.4 gm	0.28 mg	12 mg	0.54 g	11 mg
**Potassium**	**Vitamin A**	**Vitamin B1**	**Vitamin B2**	**Vitamin C**	**Vitamin B3**	**Vitamin B6**	**Zinc**
150 mg	130 I.U	0.079 mg	0.031 mg	24 mg	0.489 mg	0.110 mg	0.10 g

## Data Availability

Not applicable.

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
