# Peer review of "Application of Pineapple Waste to the Removal of Toxic Contaminants: A Review"

_toxics, 2022, doi:10.3390/toxics10100561_

Round 1

Reviewer 1 Report

This manuscript summarizes the research on the removal of heavy metals, inorganic and organic pollutants from water by nanomaterials prepared from pineapple wastes. In general, the research topic is interesting and the content is logical. This manuscript has great positive significances for promoting research in this field and can be accepted after revised. The following comments are provided as follows to improve the manuscript.

1. Sections 2.2 and 2.3 are important and attractive in this manuscript. However, the content of these sections seems to be not more than enough. Could the content be appropriately supplemented?

2. Could the authors show some data or charts from literatures to confirm that the material prepared from pineapple waste is a "Nanomaterial"?

3. The authors need to supplement some figures or data curves, such as the microscopic photos or adsorption curves of nanomaterials, which help to understand some phenomena and conclusions. These images should be obtained from the literatures which were cited in this manuscript.

4. In Section 2.6, the authors fully summarize the “Future trend” of pineapple waste. Could you supplement some explanations on the "Challenge"?

5. It is obviously inappropriate for the authors to juxtapose "Taiwan" with China in the Section 2. Taiwan is a part of the People's Republic of China. This is a common sense recognized by the international community. "Taiwan" is not a sovereign state and should not be placed in a parallel position with China, Indonesia, Malaysia, South Africa, Costa Rica, Nigeria, Philippines and Thailand. The author should change it carefully.

Author Response

Dear reviewer, thank you for your time taken and the comments brought up that have help us to improve the paper. Please find attached of the corrections done as per your comments. We hope to meet your expectations this time.

Reviewer 2 Report

This manuscript relates some aspects to the use of pineapple wastes in the depollution of wastewaters.

The authors have to improve their manuscript in order to be accepted for publication in Toxics.

1. The authors have to use L not l for volume measure.

2. In the manuscript there are some orthographic mistakes.

3. Line 96, the word "peel" is twice.

4. In title of 2.3 section the word "expulsion" is incorrect used. The authors have to change with another word.

5. Line 223 - without underline.

6. The title of the manuscript is about the removal of HM, and other pollutants, but the authors mention in this work the energy production from pineapple wastes, antioxidants and therapeutic activity etc. These application is not in agreement with the title of the manuscript.

7. Some references are wrong written (without the name of the journal). Please revise carefully this section.

Author Response

(The authors gave the same response as above.)

Reviewer 3 Report

In this manuscript, the author has reported the nanomaterials prepared from pineapple leaves for the removal of heavy metals, inorganic and organic pollutants. The paper can be accepted but needs a major revision.

(1)    This work reported numerous methods for the preparation of adsorbed nanomaterials via pineapple leaves. Comparison of advantages and disadvantages of these approaches should be provided. The application prospect of these prepared nanomaterials should be further discussed.

(2)    Please make sure your conclusions' section underscores the scientific value-added of your paper, and/or the applicability of your results. Please highlight the novelty of the review. Clearly discuss what the previous studies that you are referring to are. What are the Research Gaps/Contributions?

(3)   Discussions and conclusions should go deeper, it would be more interesting if the authors focus more on the significance of their findings regarding the importance of the interrelationship between the obtained results and sustainable development. The effective way for improving the observed situation should be further discussed.

Author Response

(The authors gave the same response as above.)

Reviewer 4 Report

Dear Authors,

Please consider the following comments before the publication of you manuscript:

  • References should be included in the manuscript following the MDPI standard. Please, modify all the references accordingly.
  • Line 79. Which statistics? Please add reference. 
  • Being a review article, several previous studies existing in scientific literature are summarized in the manuscript (pp. 1-5). Besides, the application of pineapple wastes are also analyzed and summarized (pp. 6-9). However, the objective of this study is not clear.  Please extend action 3 “Conclusion”.
  • Once I have read the manuscript, I think the title should be changed so it refers clearly to a review article. I suggest a title similar to “Application of pineapple waste to the removal of toxic contaminants: A review”

Author Response

(The authors gave the same response as above.)

Reviewer 5 Report

This review describes interesting specific work on wide applications of pineapple wastes. I regard that this review serves development of effective utilization of various plant wastes.

Title. The title is focusing on leaves while this review covers applications of other parts. I accordingly suggest changing the title to include other parts.

L 51-58. The introduction claims the significant performance of activated carbons over others. I agree that activated carbons are one of the best due to the cost effectiveness and high capacity. But others have their specific advantages like selectivity. The advantages and limitations should be fairly described.

Table 2. The alignment of columns, especially that presumably examined for multiple metals, is somehow difficult to catch. The appearance should be rearranged.

References: some references lack sufficient information.

Author Response

(The authors gave the same response as above.)

Reviewer 6 Report

The aim of the present review article was to present the possible applications of pineapple and adsorbent synthesis for the removal of pollutants.

A graphical abstract should be included.

Line 26 Authors refer to the number of deaths caused by poor water quality using data from year 2000. Please include updated data.

Line 86 ‘’The first leaf bud is lovely’’. Please consider rephrasing. May this kind of information be referenced by other scientists?

Table 1 has no labeling and there is actually no discussion on the presented information.

Table 2 has also no analysis of the presented information.

Line 182 “...presents some of the reported studies...’’ Which criteria were applies in order to present ‘’some of the reported studies’’? Please consider rephrasing.

Line 352. Nutritional value in 100 g of pineapple is labeled are Table 2, which already exists in the review.  

Authors comment that water scarcity is an important issue around the globe and state that processing of pineapple leads to the formation of wastewater. However, there is no information on the characteristics of the wastewater produced from pineapples, which should be included.

Carefully follow the instructions of the journal on the way references should be applied in the text and in the References section. In some parts of the text authors use square brackets, while in other present reference as follows; ‘’Suphnag et al. (2020)’’. The quote of references in References section is completely uneven.

https://www.mdpi.com/journal/toxics/instructions#references

Author Response

(The authors gave the same response as above.)

Round 2

Reviewer 2 Report

The authors revised the old version of the manuscript and,  now, this can be accepted.

Reviewer 6 Report

The authors addressed most concerns regarding the present manuscript and therefore it could be published in Toxics.